# Study on the Sustainable Development Factors of Agriculture-Oriented Characteristic Towns in China

Meijun Hu [1], Lei Wang [2,*], Wei Wang [3], Lingyun Tong [4] and Yuqin Lin [5]

[1] Faculty of Psychology and Educational Sciences, Vrije Universiteit Brussel, 1050 Brussels, Belgium
[2] Zhejiang Farmer Development Research Center, College of Economics and Management, Zhejiang A & F University, Hangzhou 311300, China
[3] College of Law and Political Science, Zhejiang A & F University, Hangzhou 311300, China
[4] College of Economics and Management, Zhejiang A & F University, Hangzhou 311300, China
[5] International Office (Office of Hong Kong, Macao and Taiwan Affairs), Zhejiang A & F University, Hangzhou 311300, China
* Correspondence: waleland@foxmail.com

**Abstract:** In recent years, China's agriculture-oriented characteristic towns (AOCTs) have developed rapidly. However, the development of AOCTs in different regions varies. Having examined 131 AOCTs in China, this paper constructs a comprehensive evaluation system for the sustainable development of AOCTs and applies a sub-constraint evaluation model to study the performance of AOCTs. The main findings are as follows. (1) The most important dimension for the sustainable development of AOCTs is "social development", which implies that in order to build AOCTs, the government should pay attention to social development in rural areas, especially improving medical services and public service. (2) Under the dimension of "social development", the three most important factors are the unemployment rate, per capita consumption expenditures of rural households, and per capita consumption expenditures of urban households. This implies that we must adhere to the policy for ensuring rural employment and encourage the development of local enterprises in order to provide more jobs and change their existing consumption structure. (3) Among the four main economic zones in China, the eastern region has the most advantageous factors, and the western region has the least; furthermore, regarding the disadvantaged factors, the western region has the most and the eastern region the least. This shows that AOCTs in different regions should determine their respective advantages and disadvantages, match resources accordingly, and formulate their own development strategy, which could also contribute to decreasing the gap between the eastern and western regions. Hence, the results of this study not only clearly point out the important factors for the sustainable development of AOCTs but also make them detailed and specific in order to provide the government with targeted and highly operable suggestions.

**Keywords:** agriculture-oriented characteristic towns (AOCTs); evaluation index system; sub-constraint evaluation model; social development; sustainable development

## 1. Introduction

Urbanization is a natural part of developing society, but it may bring issues such as weak agricultural infrastructure, poor coordination of urban and rural development for deserted rural areas, and poor sustainability of the ecological environment. It is always a big challenge for developing countries, when they are experiencing urbanization, to maintain sustainable development of small towns.

Characteristic towns are one of the primary drivers in China's campaign for new-type urbanization, agricultural modernization, and ecological improvement in the 21st century [1–3]. Characteristic towns, representing a new form of urbanization, have distinctive industrial and territorial functions [1,3,4]. At present, China's characteristic towns are developing rapidly. There are 403 characteristic towns listed by the Ministry of Housing

and Urban-Rural Development of China (http://www.mohurd.gov.cn/wjfb/201708/t201 70828_233078.html, accessed on 28 August 2017).

Research on characteristic towns in China is still in the initial stages, and even less attention has been given to agriculture-oriented characteristic towns (AOCTs). When discussing characteristic towns from the perspectives of industrialization, new-type urbanization, and agriculture modernization, agglomeration means that different sized industries gather and scale [2–5]. Liveability is explained as environmental optimization and life quality amelioration [6], indicating the same idea as ecological improvement.

The existing research mainly focuses on the following: defining the concept and scientific connotation of AOCTs [7–9]; establishing planning and constructing patterns of AOCTs on the basis of summarizing the practical experience of constructing AOCTs [1,9–13]; and exploring sustainable development mechanisms and paths of AOCTs by using theories, such as industrial development theory and garden city theory [4,14–16]. Although all these existing studies have provided considerable insight into the different parts of economic and administrative management policies, they each have a narrow perspective and fail to sufficiently cover the development of AOCTs from a broad perspective. Until recently, only a few articles have studied the evaluation of AOCTs from multiple perspectives.

In view of this, this paper intends to set up a comprehensive evaluation system for the sustainable development of AOCTs. A sub-constraint evaluation model is introduced into the study of sustainable development in AOCTs, using the entropy value method for reference. According to the findings of empirical analysis, recommendations are made for policy makers with respect to increasing farmers' income, reducing unemployment rate, narrowing regional disparities, and promoting social development. This study will lay a solid foundation for the sustainable development of AOCTs.

## 2. Literature Review

### 2.1. Concept of AOCTs

According to central place theory, the concept of a town is the specific category of settlement and social realities between the city and countryside [17–19]. This concept "has for centuries been of crucial importance as economic, administrative, cultural and symbolic centres in a regional context" [20]. Small towns are increasingly being perceived as a strategic focus to enhance the rural economy and narrow the disparity between urban and rural areas [21,22]. The development of small towns has become an integral part of rural and regional development theories, such as theories of regional eco-nomic divergence, industrial restructuring theories, flexible specialization, network theory, innovative milieu theory, regulation theory, agropolitan development theory, and integrated rural development theory [23–25].

As the world continues to become more urbanized, the status and functions of small towns have been constantly upgraded. In the West, small towns such as Santa Croce and Wolfsburg in Germany, Provence and Vitre in France, Davos in Switzerland, and Greenwich, County Town in the US are all localization economies associated with local excellence, product specialization, decentralized business systems, and geographical agglomeration. They have been called specialized towns, featured towns, or characteristic towns. According to Bajracharya (1995) [20], towns that perform a single function are labelled specialized towns. Zeng and Ci (2016) [26] defined specialized towns as towns that promote, support and organize a more desired lifestyle. Wang and Zhi (2018) [27] noted that a specialized town has its own highly specialized economic niche and therefore does not compete with other towns as much. The concepts of characteristic towns and feature towns are more often used by Chinese scholars. "Featured towns are defined as suburban spaces with distinct features/industries, including the following: (1) a specialized agglomeration of industry or strong potential for such a kind; (2) a greater balance between employment land use, residential and urban amenities, contrary to what is typically found in traditional industrial parks or economic development zones; (3) a strong emphasis on ecological/environmental protection" [2,7]. The concept of characteristic towns in China was first introduced by

the Zhejiang Provincial Government in 2014. The concept first appeared in the national government work report in 2017 [7]. In practice, a characteristic town, which is either a non-administrative division or inter-administrative division, is an innovative platform integrating characteristic industries, culture, tourism, and community functions [7]. Characteristic towns have distinctive industrial and territorial functions, representing a new form of urbanization, an integration of agglomeration and liveability [7]. In this study, we will use the term characteristic towns instead of other similar terms.

Currently, characteristic towns can be divided into the following three categories:

(1)  Agriculture-oriented characteristic towns (AOCTs), including three subcategories: agricultural experience towns, agricultural processing towns, and agricultural technology towns [28];

(2)  Manufacturing-oriented characteristic towns, including three subcategories: technology manufacturing towns, intelligent technology towns, and high-end manufacturing towns [28]; and

(3)  Service-oriented characteristic towns, including six subcategories: financial towns, information industry towns, medical and health towns, culture and tourism towns, sports towns, commerce towns and logistics towns [28].

Wu et al. (2017) [7] and Yang and Hao (2017) [9] defined AOCTs as a new agricultural innovation and entrepreneurship platform based on the regional characteristic agricultural resources. By integrating these resources, the integration of agriculture, industry, and the service sector are achieved, and various functions are appropriately developed. From the literal meanings of small towns, Xu et al. (2016) [13], Liu et al. (2017) [11], Yang and Hao (2017) [9], and Li et al. (2018) [10] interpreted AOCTs as a platform with distinct agricultural characteristics, complete agricultural functions, enjoyable ecological environment, and a well-developed public service system, which is "liveable and suitable for entrepreneurship, health and entertainment". To summarize, AOCTs have been defined as agricultural development platforms, covering all characteristic agricultural resources in the region.

*2.2. Theoretical Basis of the Sustainable Development of AOCTs*

The concept of sustainable development was described in the 1987 Brundtland Commission Report as "development that meets the needs of the present without compromising the ability of future generations to meet their own needs" [29,30]. The traditional development model of small towns is resource-intensive and labour-oriented, which neglects the improvement of infrastructure and the protection of the ecological environment [6,31,32]. The development model of AOCTs is quite different from that of AOCTs, which pay more attention to the exploration and cultivation of leading industries with both regional characteristics and sustainable development; mainly AOCT provides the direction for the transformation of the traditional development model of small towns [1,4,10]. Therefore, as a new type of small town, AOCTs are developed not only to meet the requirements of sustainable development but also to contribute to a new form of small towns in rural areas. In this paper, we study AOCTs by focusing on sustainable development issues.

*2.3. Sustainable Development Factors of AOCTs*

An increasing number of academic studies have focused on using composite indicators to evaluate the sustainable development factors of small towns. The sustainable development of AOCTs emphasizes people's livelihoods, social development, and quality. It advocates an urbanized lifestyle based on human-centred, highly intensive industrial development, and green intelligence. Each connotation can be interpreted from three perspectives: economic urbanization, social urbanization, and land urbanization [33,34]. The evaluation index system in the "eco-city plan" funded by the European Union covers various standards such as the urban structure, transportation, energy, material flow, and social economy [35–37]. As far as the international community is concerned, the UN Commission on Sustainable Development (UNCSD) assessed eco-cities using four criteria: society, the economy, the environment, and the system. Some scholars also analysed the three sub-

systems of urban nature, economy, and society through three indexes, namely, the natural ecological index, the economic ecological index, and the social ecological index [37–40]. The agricultural modernization index system mostly started from the dimensions of agricultural production input, agricultural comprehensive output, the rural social economy, and agricultural sustainable development [41–43].

To summarize, economic development, social development, and ecological development are indicators for the high-frequency evaluation of small towns sustainable development in authoritative literature and government reports. Therefore, this study selects three commonly used dimensions to evaluate the sustainable development factors of AOCTs. The details are shown in Table 1 below.

Economic development directly reflects the level and speed of town development and is the fundamental driving force. Therefore, the economic development quality of AOCTs should include two parts: an economic sustainability or development level and agricultural production. Most scholars use per capita GDP, the economic growth rate, the urbanization rate, and other indicators to reflect the strength of sustainable economic development [33,44]. Furthermore, agricultural modernization also reflects the level of economic sustainable development [41], and indicators such as agricultural production conditions and agricultural output efficiency are applied in this study to measure it. In addition, the growth of the town's characteristic industry is the pillar of the sustainable development of a characteristic town. Therefore, the efficiency index of characteristic industries is used in this study.

The degree of social development reflects the level of services and security available in cities and towns and reflects the degree of social civilization. Sustainable social development can be reflected by infrastructure, public services, urban and rural overall planning, and social security [16]. Ye (2001) [45] applied the notion of social well-being to show that sustainable development of small towns should mean the maintenance of the productive base of an economy relative to its population. In another book, he explored the measures of quality of life and identified the relationship between human well-being and the natural environment. Casey (2003) [46] reviewed the literature on how the clean development mechanism contributed to the sustainable development of small towns, including poverty alleviation. Dong et al. (2014) [47] discussed the role of sustainable development in small town public administration planning and in promoting social justice, equality, and citizen inclusion. Relevant indicators are the coverage rate of medical insurance, the unemployment rate, the number of technical health personnel per 10,000 people, and the number of medical beds per 10,000 people.

In addition, the construction of AOCTs is undertaken to coordinate urban and rural development and narrow the gap between urban and rural living standards. Duan et al. (2001) [36], Erlinda et al. (2016) [48], and Wurst (2020) [49] studied sustainable urban development issues. Marek and Ada (2021) [50] focused on the sustainable tourism sector in small towns. Other scholars such as Wang et al. (2011) [5], Li (2017) [51], and Wurst (2020) [49] addressed the theory and implications of sustainable construction. Therefore, when evaluating the social sustainable development level of AOCTs, we should not only examine their external development but also consider their internal differences. Indexes such as the income ratio of urban and rural residents, the per capita consumption rate and per capita disposable income of urban residents, the per capita net income of farmers, and the per capita consumption expenditures of rural residents are quoted in this study.

In recent years, the importance of environmental quality has been well recognized and integrated into the evaluation index system of urban sustainable development, including the aspects of urban greening and pollution control. This paper selected the per capita green park area, forest coverage rate, harmless treatment rate of domestic waste, and energy consumption reduction rate per CNY 10,000 to evaluate the ecological environment quality of AOCTs.

**Table 1.** Evaluation index system of development factors of AOCTs.

| Dimension | Sub-Dimension | Factor |
|---|---|---|
| A. Economic Development | $A_1$ Economic level | $A_{11}$ Per capita GDP<br>$A_{12}$ Economic growth rate<br>$A_{13}$ Urbanization rate |
| | $A_2$ Characteristic industry efficiency | $A_{21}$ Proportion of characteristic industry investment with respect to total investment<br>$A_{22}$ Proportion of business income of characteristic industry service industry with respect to that of the town service industry<br>$A_{23}$ Proportion of the total industrial output value of characteristic industries with respect to the total industrial output value of small towns |
| | $A_3$ Agricultural production conditions | $A_{31}$ High standard farmland proportion<br>$A_{32}$ Agricultural mechanization rate<br>$A_{33}$ The proportion of agricultural scientific research investment with respect to GDP |
| | $A_4$ Agricultural output efficiency | $A_{41}$ Land output rate<br>$A_{42}$ Yearly growth rate of primary industry<br>$A_{43}$ Annual growth rate of agriculture, forestry, livestock, fisheries |
| B. Social Development | $B_1$ Living standards | $B_{11}$ Medical coverage<br>$B_{12}$ Unemployment rate |
| | $B_2$ Public service | $B_{21}$ Number of health technicians per 10,000 people<br>$B_{22}$ Number of medical beds available to 10,000 people |
| | $B_3$ Urban and rural development | $B_{31}$ Income ratio of urban and rural residents<br>$B_{32}$ Per capita consumption expenditures of urban households<br>$B_{33}$ Per capita disposable income of urban residents |
| | $B_4$ Rural development | $B_{41}$ Per capita disposable income of rural residents<br>$B_{42}$ Per capita consumption expenditures of rural households |
| C. Ecological Development | $C_1$ Urban greening | $C_{11}$ Per capita green park space area<br>$C_{12}$ Forest coverage |
| | $C_2$ Pollution control | $C_{21}$ Harmless treatment rate of domestic waste<br>$C_{22}$ Energy consumption reduction rate per CNY 10,000 |

Note: Except for indicators such as the rate of unemployment and income ratio of urban and rural residents, all other indicators are positive. Major Sources: Stevenson and Lee, 2001 [43]; United Nations Human Habitat, 2002 [30]; Zhou et al., 2014 [52]; Wu et al., 2016 [8]; Liu et al., 2017 [53]; Key Indicators Database (KIDB): https://kidb.adb.org/kidb/, accessed on 24 August 2022.

## 3. Data and Research Methodology

### 3.1. Sample Selection and Data Resources

In regard to sample selection, AOCTs are administrative units. To avoid the impact of regional differences to ensure the accuracy of the results, this paper chose samples with typical representation by covering the 131 AOCTs on the list of first and second batches of AOCTs published by the Ministry of Housing and Urban-Rural Development.

The data were collected from the following resources: the *China County Statistical Yearbook 2017 (Township Volume)*, county-level websites and municipal-level statistics bureaus, published statistical bulletins of all districts and counties in 2017, local government planning documents, and official AOCT websites.

### 3.2. Determination of the Rationality of Index System Construction

The basis for judging whether the index system is reasonable lies in the amount of index information content, which is expressed by data variance through factor analysis.

The covariance matrix of the index data is set as *S*; and the trace of the covariance matrix is set as *trS*, which represents the sum of the variances of the indexes on the principal diagonal line of the covariance matrix. *S* is the number of indexes after selection, and *H* is the number of selected indexes. In denotes the information contribution rate of the index to the original index after selection [54].

If less than 30% of the original index information can be used to reflect more than 90% of the original index information in the process of building the index system, the

construction of the index system is considered to be successful. After calculating the original data obtained in this study, the information contribution rate is obtained.

$$In = \frac{trS_s}{trS_h} = \frac{4636.375817}{5011.630719} = 92.5\%$$

In this study, 28.74% (25/87) of the original indexes approximately reflect the information of 92.5% (>90%) of the original indexes, which proves that the index system established is reasonable.

*3.3. Sub-Constraint Evaluation Model Formula*

3.3.1. Determining the Combination Weight of Indexes

The subordination weight of index $j$ of town $i$ is recorded as $f_{ij}$, and the score value of index $j$ of town $i$ is given as $P_{ij}$. N is the number of towns evaluated, m is the number of indexes, and $H_j$ is the entropy value of evaluation index $j$. The calculation formula of the entropy weight of the evaluation index is as follows, satisfying the following [55–57]

$$\sum_{j=1}^{m} W_j = 1.$$
$$W_j = \frac{1-H_j}{m-\sum_{j=1}^{m} H_j} \tag{1}$$

In the formula: $f_{ij} = \frac{P_{ij}}{\sum_{j=1}^{n} P_{ij}}$; $H_j = -\frac{1}{lnn} \sum_{j=1}^{n} f_{ij} ln f_{ij}$

3.3.2. Defining the Degree of Optimal and Secondary Subordination and Sub-Constraints

As for evaluation index $j$ with the best subordination [58], $V^\alpha = \left(v_1^\alpha, v_2^\alpha, \ldots, v_m^\alpha\right) = (1, 1, \ldots, 1)^T$, which means that for all small towns, the value is 1. As for evaluation index $j$ with the worst subordination, $V^\beta = \left(v_1^\beta, v_2^\beta, \ldots, v_m^\beta\right) = (0, 0, \ldots, 0)^T$, which means that for all small towns, the value is 0.

Among $m$ evaluation indexes, $t$ evaluation indexes are selected randomly $(1 \leq t \leq m)$, and the vector composed of the worst score value of the t evaluation indexes among N evaluation towns is the sub-constraint. The sub-constraints consisting of $t$ indexes are set as $V_{sk}, v_{sk} = (v_{sk1}, v_{sk2}, \ldots, v_{im})^T = (0, 0, \ldots, 0)^T$ [58,59].

3.3.3. Determining Weight Quality

For town $i$, the Euclidean weight distance $d(V_i, V^\alpha)$ between the subordinate degree vector $V_i$ and the optimal subordinate degree $V^\alpha$ is expressed as [58–62]

$$d(v_i, v^\alpha) = \left[\sum_{j=1}^{m} w_j \left(v_{ij}, v_j^\alpha\right)^2\right]^{1/2} \tag{2}$$

If the subordinate degree of town $i$ to the optimal vector $R^\alpha$ is expressed as $u_i^\alpha$, the weight quality of town i to the optimal vector $R^\alpha$ is expressed as [58–60]

$$D(V_i, V^\alpha) = u_i^\alpha d(V_i, V^\alpha) \tag{3}$$

3.3.4. Determining Weight Diversity

The Euclidean weight distance between the subordinate degree vector $V_i$ and the secondary subordinate degree vector $R^\beta$ for town I is expressed as $d(v_i, v^\beta)$ [58]. The calculation formula is as follows (2).

If the subordinate degree of town $i$ to sub-vector $R^\beta$ is expressed as $u_i^\beta$, the weight diversity of town i to the optimal vector is expressed as $D(v_i, v^\beta)$ [58,60,63]. The calculation formula is as follows (3).

The Euclidean weight distance between the subordinate degree vector $V_i$ and the sub-constraint $V_{sk}$ for town i is expressed as [59,60,64]

$$d(v_i, v_{sk}) = \left[ \sum_{j \in sk} \left[ \frac{w_j}{\sum_{j \in sk} w_j} (v_{ij} - v_{sk})^2 \right] \right]^{1/2} \qquad (4)$$

If the subordinate degree of town $i$ to sub-constraint $S_k$ is expressed as $u_{isk}^\beta$, the weight diversity of town i to sub-constraint $S_k$ is expressed as [58–60]

$$D(v_i, v_{sk}) = u_{isk}^\beta d(v_i, v_{sk}) \qquad (5)$$

3.3.5. Establishing Evaluation Model

The objective function of the model is determined by the minimum value of the weighted sum of squares to the weight quality and the weighted diversity of each town. On this basis, a comprehensive evaluation model based on sub-constraints is established [58–60].

$$minF = \sum_{i=1}^{m} \left[ \frac{1}{L} \left( D\left(v_i, v^\beta\right)^2 + \left(D(v_i, v^\alpha)^2\right) + \sum_{k=1}^{L-1} \left(D(v_i, v_{sk})^2\right) \right) \right] \qquad (6)$$

The optimal solution of the model can be obtained, and the optimal solution of town evaluation can introduce sub-constraints [52].

$$u_{is}^\alpha = \frac{Ld\left(v_i, v^\beta\right)}{Ld^2\left(v_i, v^\alpha\right) + Ld^2\left(v_i, v^\beta\right) + d^2(v_i, v^\alpha)d^2\left(v_i, v^\beta\right) \sum_{k=1}^{L-1} \frac{1}{d^2(v_i, v_{sk})}} \qquad (7)$$

## 4. Results

*4.1. Comprehensive Evaluation of the Influencing Factors of the Sustainable Development of AOCTs*

4.1.1. Determining Key Dimension

According to the establishment and solution process of the comprehensive evaluation model for the sustainable development of AOCTs, as mentioned above, the evaluation results of 131 AOCTs with no sub-constraints and with each dimension level as sub-constraints shown are obtained. Based on the subordinate degree of each town with or without sub-constraints, each town is ranked.

Through longitudinal calculation, the difference between the ranking of sub-constraints and non-sub-constraints of each small town under different conditions is obtained, and the sum is calculated to symbolize absolute values of the differences between towns under each main target and dimension. The larger the algebraic sum of the main target is, the greater the influence of the dimension on the development of small towns. In other words, the main target and dimension have greater significance.

A presentation of the outcome sub-constraint model is illustrated in Table 2. The table shows that the most important dimension for the sustainable development of the studied AOCTs is the "social development" dimension, which gives strong evidence to Zhang's (2013) [16] theory that the level of social well-being and access to services are precursors to improving livelihoods and are enabling factors of people's engagement in productive activities. Furthermore, this result is more accurate than the conclusion of Yang and Hao (2017) [9] that "an important factor in building AOCT is economic and social development".

4.1.2. Determining the Main Factors

Table 2 shows that the "social development" dimension layer is the key dimension layer for the sustainable development of AOCTs. Therefore, the nine factors under this dimension layer are analysed to determine the key factors that affect the social development of AOCTs. The situation is shown in Table 3.

**Table 2.** Comprehensive results of sub-constrained subordination rankings.

| Dimension | Maximum Value | Minimum Value | Sum of Absolute Values |
|---|---|---|---|
| Economic Development | +130 | −130 | 8512 |
| Social Development | +130 | −129 | 8576 |
| Ecological Development | +129 | −128 | 8384 |

Note: "+" and "−" represent the change of the ranking position of the same town with or without sub-constraints; "+" means that the ranking position of the town increases after adding sub-constraints group, and "−" means that the ranking position of the town decreases after adding sub-constraints group.

**Table 3.** Comprehensive results of ranking changes in different factors.

| Serial Number | Factor | Maximum Value | Minimum Value | Sum of Absolute Values |
|---|---|---|---|---|
| 1 | Unemployment rate | +130 | −130 | 8580 |
| 2 | Per capita consumption expenditures of rural households | +129 | −126 | 7796 |
| 3 | Per capita consumption expenditures of urban households | +122 | −126 | 7606 |
| 4 | Per capita disposable income of rural residents | +128 | −127 | 7208 |
| 5 | Number of health technicians per 10,000 people | +129 | −130 | 6900 |
| 6 | Per capita disposable income of urban residents | +127 | −106 | 6834 |
| 7 | Income ratio of urban and rural residents | +128 | −106 | 6752 |
| 8 | Number of medical beds available to 10,000 people | +126 | −113 | 6592 |
| 9 | Medical coverage | +130 | −112 | 5334 |

Note: "+" and "−" represent the change of the ranking position of the same town with or without sub-constraints; "+" means that the ranking position of the town increases after adding sub-constraints group, and "−" means that the ranking position of the town decreases after adding sub-constraints group.

Table 3 shows that among the nine factors under the most important dimension level of "social development" in the sustainable development indicator system of AOCTs, the three most critical factors are the "unemployment rate", "per capita consumption expenditures of rural households", and "per capita consumption expenditures of urban households". This is consistent with Jan and Milan (2007) [65], Aedín et al. (2018) [66] and Popescu et al. (2021) [67], in that the reduction of unemployment is crucial to the stability of social development. On the other hand, this partially differs from Bakare (2014) [6], who proved that decrease in unemployment and increase in guaranteed income are important variables for improving urban poverty and maintaining stable social development. Therefore, the results of this study provide a more tangible guide to social sustainability and are an important basis for studying the social sustainability of towns.

*4.2. Analysis of 131 AOCTs by Regions*

In order to reveal the sustainable development characteristics of the sample towns in different regions, the distribution results of the advantage factors and disadvantage factors of 131 towns were imported into the SPSS 22.0, and the cluster analysis method combined with the average Euclidean weight distance (d) was used to analyse the cluster analysis dendrogram of 131 towns in different economic regions. Divided by d = 14, the 131 samples were aggregated into four categories: towns S10, S65, S57…S14 were grouped into category I; towns S41, S49, S42…S114 were gathered into category II; towns S56, S72, S18…S28 were gathered into category III; and towns S113, S118, S100…S45 were grouped into category IV, as shown in Figure 1. Among the categories, the first category of towns has numerous disadvantages and very few advantages. Their overall development level is generally ranked low. The second category is composed of towns with more disadvantages than advantages. Their disadvantage factors are mostly concentrated. The third category of towns has more balanced advantages and disadvantages. Their development level is ranked in the middle. The fourth type of town has numerous advantages and few disadvantages. Their development level is generally ranked high.

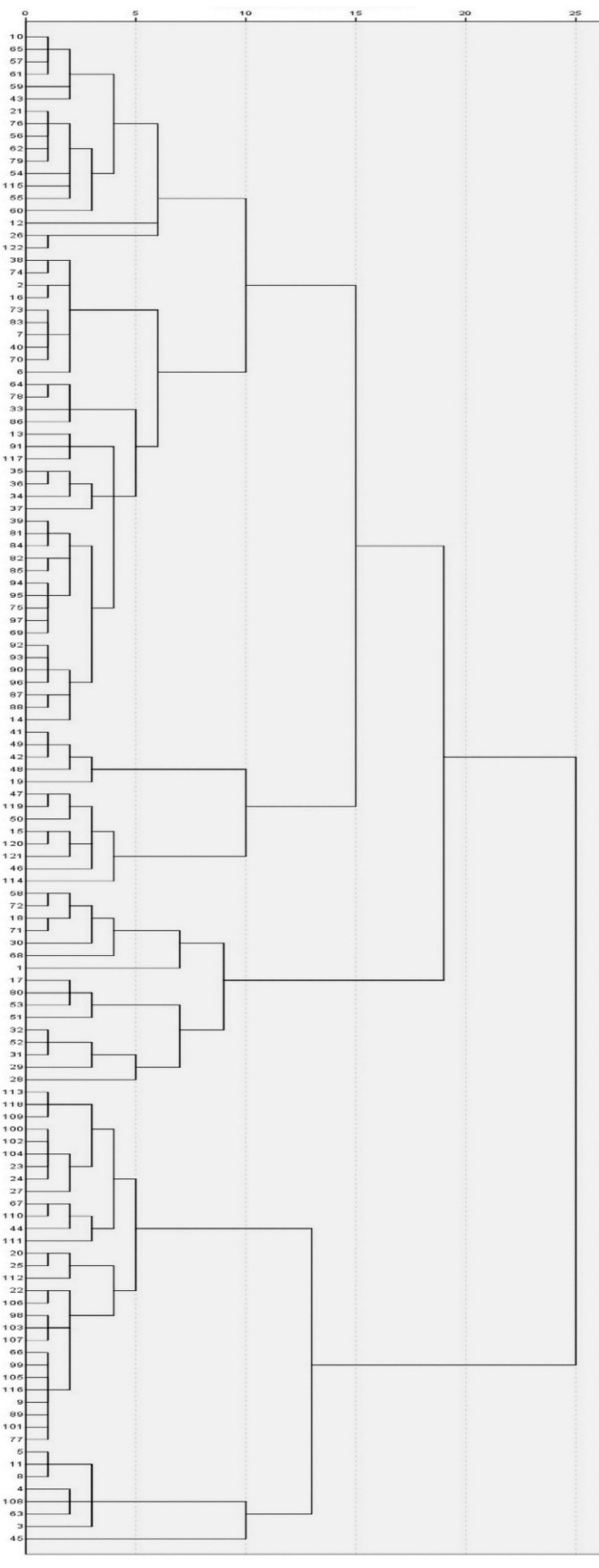

**Figure 1.** Cluster analysis dendrogram of 131 AOCTs in different economic regions.

Using the cross-analysis method to conduct descriptive analysis on the category and area of the town, we found the following: category I has a total of 53 towns, of which 21 are in the western region, accounting for 39.62%; category II has a total of 26 towns, of which 23 are in the western region, accounting for 88.46%, and there are no category II towns in the eastern region; category III has a total of 20 towns, of which 7 are in the western region, accounting for 35%; and category IV has a total of 32 towns, of which there are no fourth-class towns in the western region but 21 in the eastern region, accounting for 65.63% (as shown in Table 4). The results of the cross-analysis show that with the improvement of the small town category level, the proportion of small towns in the western region is continuously decreasing. Additionally, the AOCTs in the western region are basically at the sustainable development level of category II, that is, their current level of sustainable development is generally ranked low.

**Table 4.** Cross Analysis for Regions and Categories of towns.

| | | Regions | | | | |
| --- | --- | --- | --- | --- | --- | --- |
| | | Northeastern China | Western Region | Eastern China | China Central | Total |
| Categories of towns | Category I | 11 | 21 | 8 | 13 | 53 |
| | Category II | 2 | 23 | 0 | 1 | 26 |
| | Category III | 1 | 7 | 8 | 4 | 20 |
| | Category IV | 7 | 0 | 21 | 4 | 32 |
| Total | | 21 | 51 | 37 | 22 | 131 |

Combining the results of the cluster analysis of different economic regions with the distribution of the advantageous factors and disadvantageous factors in their respective regions, the overall situation is shown in Figure 2.

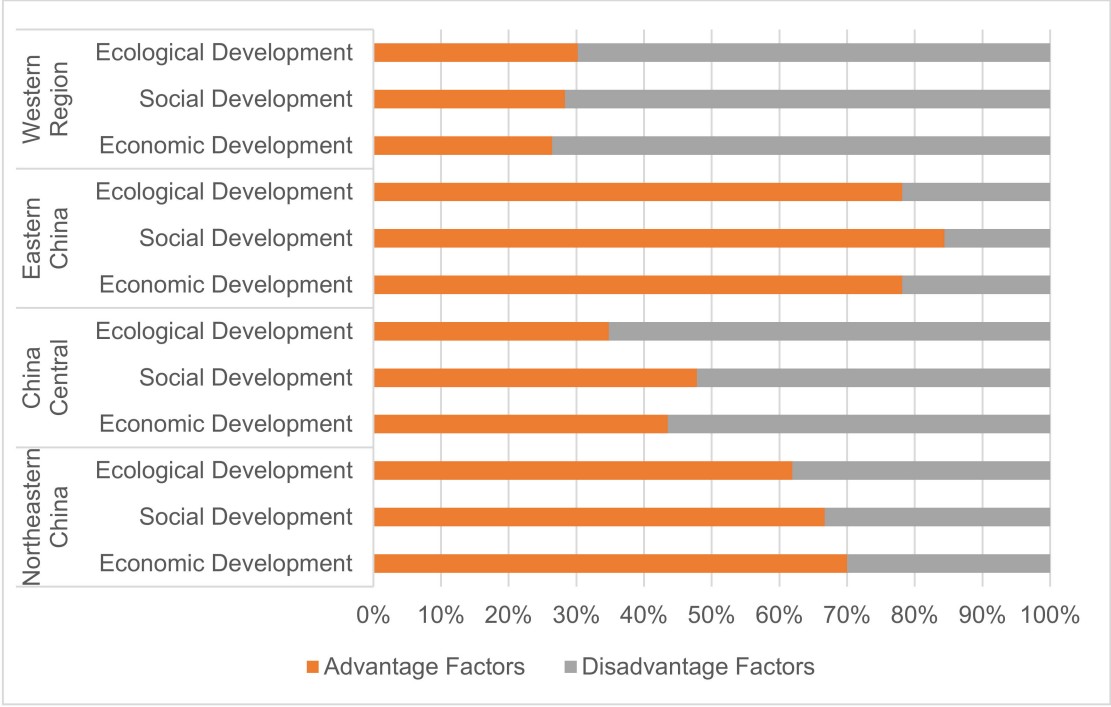

**Figure 2.** Advantageous and disadvantageous factors of the sustainable development of AOCTs in different regions.

Figure 2 shows that most of the towns in the fourth category are located in the eastern region; and the area has the highest portion of the advantageous factors, among

which the dimension of "social development" develops best. In contrast, most of the first-type towns are located in the western region, and the weight of disadvantageous factors is heavy in the region, among which the dimension of "economic development" develops the least. The results concord with Pike et al. (2007) [68], Garretsen et al. (2013) [69], Barca et al. (2012) [70], and Qiu and Wang (2022) [71] and propose that regional inequality increased quite markedly within countries, with poorer regions lagging further behind the highest income regions. The results of this study show that despite its low level of social development, the western region is in the second category of sustainable development with many disadvantageous factors and few advantageous factors. This is consistent with Wang et al. (2018) [27].

## 5. Discussion and Conclusions

The empirical results of the study make important contributions to the sustainable development of AOCTs in China and the sustainable development of small towns in general. This paper uses the entropy value method and sub-constraint evaluation model to measure the sustainable development of AOCTs. It is found that "social development" is the most important dimension for the sustainable development of AOCTs. The study also found that the unemployment rate, per capita consumption expenditures of rural households, and per capita consumption expenditures of urban households are the three most important factors affecting social development. In addition, we discovered that the eastern region of China's four main economic zones has the most advantageous factors while the western region has the least; in addition, the western region has the most disadvantaged factors while the eastern region has the least.

First, the study found that a significant dimension for the sustainable development of AOCTs is "social development". The availability of essential medical and health resources, and the standard of local public health care need improvement. Especially in the field of rural medical care, boosting rural governance is crucial for rural vitalization. Therefore, rural healthcare providers can pursue multifaceted strategies to increase the availability, accessibility, and affordability of health care. For example, the local government should consider establishing a comprehensive system of public health services, including disease prevention and control, health education, and health supervision. Community-level medical and healthcare system healthcare networks that provide public health services are also essential [67]. In addition, an excellent medical security system is necessary for the sustainable development of a small town. This system should comprise basic medical insurance covering urban and rural residents, supplemented by medical insurance and commercial health insurance in various forms [72]. In order to encourage social development, we also should continuously raise the quality of public services, enhance the delivery system, and thoughtfully plan and organize all categories of rural public services. According to regional difference, it is required to enhance the alignment and efficiency of public financial expenditure and economic and social growth, as well as provide support for rural public services in the western and north-eastern regions.

Second, we identified the effective factors for sustainable development in AOCTs. The per capita consumption expenditures of rural households and the per capita consumption expenditures of urban households are critical factors, whereas the unemployment rate is the major challenge. Therefore, national and local governments should formulate and implement an active employment policy in accordance with the actual situation of these towns. For example, national governments could set up special funds to support rural workers in receiving professional vocational training. Governments at all levels should enact preferential tax policies to support the re-employment and entrepreneurship of the unemployed [65,66]. In addition, local governments should take the development of community services, catering, trade circulation, tourism, and other tertiary industries as the major orientation to expand employment, to create more employment opportunities for farmers to increase their income.

Furthermore, the study showed that the best performing AOCTs are those in the eastern region, and the worst ones are in the western region. According to the data analysis, it can be seen that the level of social well-being of the eastern region is rather high, and most of them have successfully transitioned from development led by economic growth to sustainable development. Based on this situation, the national government could actively promote "east-west pairing-off cooperation" policies that link more developed eastern regions and less developed western regions at different levels of economic growth for more equitable development and resource distribution. In other words, the government should mobilize eastern regional assets and exploit synergies to improve infrastructure in the western region. This involves industrial cooperation, human resource exchanges, financial assistance, and ecological and environmental problem solving. For instance, we propose to create a business- and investment-friendly environment in the western regions to attract large firms to territories with a weak industrial region. In addition, the government may establish an agricultural processing centre in order to attract other types of industries to locate nearby. Through an integrated local development approach, the development of existing agricultural areas in less-developed regions aims to reduce imbalances between economic sectors and between developed and underdeveloped regions. As for the eastern region, we recommend to enhance investment promotion measures such as one-stop service for investment, high-quality infrastructure construction (e.g., railways, roads, electricity and telecommunications, and cooperation between agriculture industries), facilitation of the mass transportation of agricultural commodities, deep processing and cold chain logistics, and management of agricultural products.

Finally, the governments of the two regions should also enhance industrial human resource development. That is, the eastern region should provide experts in related fields and advanced technical support to help the western region implement training programmes in supporting industries, small and medium-sized enterprises, and the agricultural and food industry. At the same time, based on the region's resource endowment and industrial base, the western region should stimulate the enthusiasm of leading enterprises in the eastern region to invest in the western region. The local government in the western region should also support the training of a number of poor people with high participation in characteristic industrial bases from the eastern region to introduce a number of labour-intensive enterprises and cultural tourism enterprises that can provide more jobs, and promote the industrial development of the western region to enhance economic vitality. Over the past years, the western region has taken resource development as the primary task, neglecting the basic starting point of large-scale development of local ecological environment protection. We believe that the authorities should not only formulate appropriate resource taxes according to the degree of threat to the local environment, but also establish a reasonable East-West cooperation in resource development projects at the same time, strengthening the ecological construction and environmental protection of the capital investment.

The advantages of this study are that, for the sustainable development of AOCTs, it establishes a comprehensive evaluation index system, points out the most important dimension, and evaluates those important factors. Based on the results, specific, detailed recommendations are provided. Moreover, our research combines a sub-constraint evaluation model with a cluster analysis method and a cross-analysis method. This is creative for researches related to sustainable development and regional differences in the town.

The limitation of this study is that the model of this study was not optimized. Efforts have been made to simplify the computational process of the sub-constraint comprehensive evaluation model, and some well-known computational processes are described briefly. However, it is not easy to simplify the calculation part in the model construction, and the rationality of the partial description of the simplified calculation process needs further discussion. Of course, in a follow-up study, we can adopt a variety of research methods with different modelling. Above all, we can identify tests to ensure the effectiveness and feasibility of the evaluation model in the hope of solving any relevant problems.

**Author Contributions:** Data curation, M.H., L.T. and Y.L.; Writing—original draft, M.H.; Writing—review & editing, L.W. and W.W. All authors have read and agreed to the published version of the manuscript.

**Funding:** This research was funded by The National Social Science Fund of China, grant number: 19BGL174.

**Institutional Review Board Statement:** Not applicable.

**Informed Consent Statement:** Not applicable.

**Data Availability Statement:** Not applicable.

**Conflicts of Interest:** The authors declare no conflict of interest.

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
