# Peer review of "Study on the Sustainable Development Factors of Agriculture-Oriented Characteristic Towns in China"

_sustainability, doi:10.3390/su141912292_

Round 1

Reviewer 1 Report

The paper is well structured and also well positioned in the scope of rural town development. 

The readability  is good and the retionalle of the proposal is clear in the work.

About analytical structure: it not new in techniques but the case study (chines AOCTs) is relevant.

I ask to change the title in order to refer clearly on the case study: it is a research connected with China and the reader has to clearly identify it since from the beginning.

The bibliography should be improved including relevant works on the topic widely available at global level.

I suggest minor revisions

Author Response

Dear Reviewer,

Thank you very much for your time and kind recommendation.

Based on your comments, we have completed the revisions point by point and given explanations. Please see the attachment.

Kind regards,

Meijun Hu

Reviewer 2 Report

the article is innovative,
interestingly written,
It can be published as presented.

Author Response

Dear Reviewer:

Thank you very much for your time and kind recommendation.

Kind regards,

Meijun Hu

Reviewer 3 Report

The study aims at providing a comprehensive evaluation index system of the sustainable development factors of AOCTs and investigate advantageous factors and disadvantageous factors from a regional development perspective. In my opinion the paper is really interesting and worthy to be published after a few changes.

In the abstract, for better consistency, since there is no indicator for social security in the paper, I suggest deleting the reference to this item. I also suggest replacing "employment rate" with "unemployment rate" (like the factor on pages 5 and 9).

Some concepts mentioned in the text require further and more up-to-date bibliographical references. I am referring to: central place theory (line 38); sustainable development (line 126); “eco-city plan” (line 145; please see the new EU Policy on the Urban Environment at the link: https://ec.europa.eu/environment/urban/tool.htm); entropy value (line 238), for which you might consider, for instance,“A Multi-Level Fuzzy Comprehensive Evaluation Method for Knowledge Transfer Efficiency in Innovation Cluster” by Xiaoli Zhang and Rui Xu; Hindawi Mobile Information Systems Volume 2022, Article ID 3949597, 12 pages; https://doi.org/10.1155/2022/3949597.

 As far as methodology is concerned, I wonder why the list of factors on page 5 does not include demographic aspects such as problems of aging and rural depopulation,... Maybe is it because of gaps in the statistical sources on page 6?

Discussion and conclusions sections are not adequately connected with the results and the entire text. For instance, there are no previous paragraphs which have developed the discussion on rural social security (please see lines 364-376). Furthermore, the long list of proposals to governments for a more balanced regional development should be better linked to what is best expressed in the abstract regarding the importance of a national coordination strategy and different place-based policies (one of the best references in this respect is “The case for regional development intervention: place-based versus place-neutral approaches” by Fabrizio Barca, Philip McCann, Andrés Rodríguez-Pose; 2012; https://doi.org/10.1111/j.1467-9787.2011.00756.x).

Finally, I realise that the evaluation model applied required  a great effort of simplification, but an additional effort in communication could benefit the usefulness of the evaluation system in policy decision-making processes.

I also suggest checking: a) the definition of certain factors in the table 1 (e.g. A22); b) the bibliography in terms of order and completeness against the text (e.g. Fang et al, 2021 is missing); c) the caption of the figure on page 11.

Author Response

Dear Reviewer,

Thank you very much for your time and kind recommendation.

Based on your comments, we have completed the revisions point by point and given explanations.Please see the attachment.

Kind regards,

Meijun Hu

Reviewer 4 Report

1. The abstract is too long, it is recommended to compress it.

2. It is recommended to separate the introduction and literature review, describe in detail the main differences between this paper and previous literature, and explain the theoretical value and application value of this paper.

3. The theoretical background and the methods section are combined.

4. Edit formulas and letters with a professional editor.

5. The discussion part and the results part should be combined. The differences between the results of this paper and those of previous scholars should be discussed, and the contributions of this paper should be highlighted.

6. There are too many references in Chinese. It is recommended to increase the literature of foreign authors to reflect the research of scholars from different countries in this field and meet the requirements of international journals.

7. Update some literatures, and a large number of literatures are outdated.

Author Response

(The authors gave the same response as above.)

Round 2

Reviewer 4 Report

Accept in present form